# Precise determination of molecular adsorption geometries by room temperature non-contact atomic force microscopy

Timothy Brown [1✉], Philip James Blowey [1] & Adam Sweetman [1✉]

High resolution force measurements of molecules on surfaces, in non-contact atomic force microscopy, are often only performed at cryogenic temperatures, due to needing a highly stable system, and a passivated probe tip (typically via CO-functionalisation). Here we show a reliable protocol for acquiring three-dimensional force map data over both single organic molecules and assembled islands of molecules, at room temperature. Isolated cobalt phthalocyanine and islands of $C_{60}$ are characterised with submolecular resolution, on a passivated silicon substrate (B:Si(111)-($\sqrt{3} \times \sqrt{3}$)$R30°$). Geometries of cobalt phthalocyanine are determined to a ∼10 pm accuracy. For the $C_{60}$, the protocol is sufficiently robust that areas spanning 10 nm × 10 nm are mapped, despite the difficulties of room temperature operation. These results provide a proof-of-concept for gathering high-resolution three-dimensional force maps of networks of complex, non-planar molecules on surfaces, in conditions more analogous to real-world application.

[1] The University of Leeds, Leeds, United Kingdom. ✉email: T.J.Brownphy@leeds.ac.uk; A.M.Sweetman@leeds.ac.uk

Atomic and molecular-scale surface structures are often characterised and measured using scanning probe microscopy (SPM) techniques. Scanning tunneling microscopy (STM) is capable of mapping the local density of states of a sample, by detecting a current of tunneling electrons through the junction between a metal tip and conducting surface[1]. Atomic force microscopy (AFM)[2] is principally similar to STM but detects a tip-sample force across the junction by mounting the tip on a deflecting cantilever. Each technique is sufficiently sensitive to achieve atomic resolution in imaging and spectroscopic measurements. Further, in non-contact atomic force microscopy (NC-AFM)[3], the sensitivity of AFM can be enhanced by oscillating the tip at its eigenfrequency, $f_0$ and measuring instead the change, $\Delta f$, and at a fixed amplitude, $A_0$, due to the tip-sample force-interaction.

Gross et al.[4] pioneered the use of carbon monoxide (CO) functionalised probe tips in NC-AFM, in order to resolve detailed chemical structures of molecules on metal surfaces. Currently, submolecular resolution imaging has become routine in NC-AFM, through the use of quartz tuning fork probes[5] with CO functionalised tips in low temperature (~5 K) ultra-high vacuum (UHV) systems, and has yielded enormous progress in our ability to characterize molecular and two-dimensional materials at the atomic scale. Using NC-AFM, one can observe the individual stages of chemical reactions[6], bond order[7] and charge localisation[8] within organic molecules and carry out detailed investigations into on-surface synthesised two-dimensional (2D) materials[9]. In particular, many of the results throughout the literature have made use of three-dimensional (3D) force mapping, to elucidate key information about a given system, beyond that obtained via conventional imaging, such as the precise adsorption geometry of molecules[10,11], or the degree of tip deflection during data acquisition[12].

A wide variety of 2D supramolecular architectures on metal surfaces have been well-studied throughout the literature via SPM. Through self-assembly, distributions of molecules on surfaces can arrange into well-ordered networks, governed by the molecule-molecule and surface-molecule interactions. Self-assembly has received much attention[13–15], owing to its wide range of applications, ranging from providing selective ion exchange[16–18], to preparing functional materials[19–21] and fabricating novel nanodevices[22,23]. STM is often the technique used for the study of assembled molecules, owing to its ability to resolve individual molecules, and its relative simplicity. While NC-AFM has previously been used to study intermolecular bonded system[24,25], these experiments, in addition to force mapping experiments in general, have historically been performed under cryogenic conditions. However, if a molecular sample has applications in room temperature conditions, it follows that it ought to be characterised and studied under similar conditions, in order to more accurately report its structure / behaviour.

The most significant challenge of room temperature operation of NC-AFM is the thermal drift between the tip and sample. If uncorrected, the tip-sample drift will lead to distortions in the imaging / spectroscopic data. For resolving the internal structures of molecules, it is necessary to operate in constant height mode, rather than in feedback, in order to measure both the attractive and repulsive forces, at a given height. In the case of force mapping in particular, the data acquired in constant height mode must have negligible drift to obtain an accurate reconstruction of the force field, from the ($\Delta f$) signal. This can be achieved using in-situ feedforward correction, in which an atom tracking technique[26] locks the tip onto a reference point of the surface. The calculated tip-sample drift is then compensated by applying a voltage ramp to the piezo scan-tube of the microscope[27].

We favoured an in-situ method of drift correction as opposed to post-*hoc* alignment methods[28], to better ensure that the molecular features imaged at close approach arose from genuine tip-sample interactions rather than potential artefacts of data correction[29]. In recent work by the current authors[30], submolecular resolution force mapping over a single planar organic molecule was demonstrated at room temperature, using conventional silicon cantilevers. In this paper, we demonstrate further versatility and flexibility of room temperature force mapping, through its application across complex, non-planar molecular systems and areas large enough to characterise islands of self-organised / self-assembled molecules.

Prior to experiment, the appropriate substrate was considered. For submolecular resolution, a chemically passive tip is normally required in order to approach sufficiently close to the sample that the repulsive forces can be measured, without a loss in stability. As such, our substrate needed to be semiconducting, as these can facilitate, via gentle indentation into the surface, the chemical passivation of the tip[31]. Preparation of the tip in this way can yield a molecular or semiconducting cluster termination, which can be conducive to unreactive tips and hence, chemical-bond resolution[32]. Concurrently, the molecule-substrate interaction must be sufficiently strong to inhibit molecular diffusion during imaging (which is prevalent in room temperature conditions), however it also ought to be weak enough that some amount of diffusion can occur in order to form intermolecular bonds when appropriate[33]. The Si(111)-(7 × 7) reconstructed surface, as used in the aforementioned work by the current authors, for molecular force mapping experiments[30], would not work in this instance, as the strong covalent bonding between the molecules and the surface inhibits molecular diffusion, and thus the formation of islands / ordered networks. Instead, a boron passivated Si(111)-($\sqrt{3} \times \sqrt{3}$)$R30°$) substrate (hereon abbreviated to B:Si(111)) was used, whereby the segregation of boron dopants to the surface inhibits the covalent bonding between the adatoms and the molecules (see Fig. 1a–c). This inhibition of the molecule-substrate interactions allows for the formation of assembled molecular networks, examples of which have been previously observed on the B:Si(111) surface by STM[34–36].

Preliminary force mapping experiments on the B:Si(111) substrate were carried out using single cobalt phthalocyanine (CoPc) molecules. Metal phthalocyanines are extensively studied in SPM, due in part to their applications in industrial catalysis[37,38], fabrication of organic thin film transistors[39,40] and other novel functional materials[41,42]. The boron-passivated surface allows numerous types of phthalocyanine to self-assemble according to their molecule-molecule interactions. These interactions on the B:Si(111) surface are conducive to forming tilted 2D structures for H2Pc, CuPc, and ZnPc, due to $\pi - \pi$ stacking at sub-monolayer coverages[36]. In contrast, CoPc has previously been shown to adsorb highly aligned to the substrate, with benzene rings planar to the surface, despite surface passivation[43,44]. As such, it is a suitable prototype system for submolecular imaging / mapping of single molecules.

In this study we focus on low coverages of CoPc to demonstrate the ability of room temperature NC-AFM to identify changes on the order of ~10 pm in the molecular geometry. Following that, we investigate $C_{60}$, which is well-characterised in NC-AFM throughout the literature, and is amenable to forming islands on the B:Si(111) surface[45,46]. $C_{60}$ is therefore a good prototypical system to demonstrate the applicability of the force mapping techniques for large areas of self-assembled networks. Both molecular systems were initially imaged in STM, in order to assess the approximate coverage, before imaging in NC-AFM (Fig. 1e, f, k, & l). For the CoPc, the relative tilt and buckling of

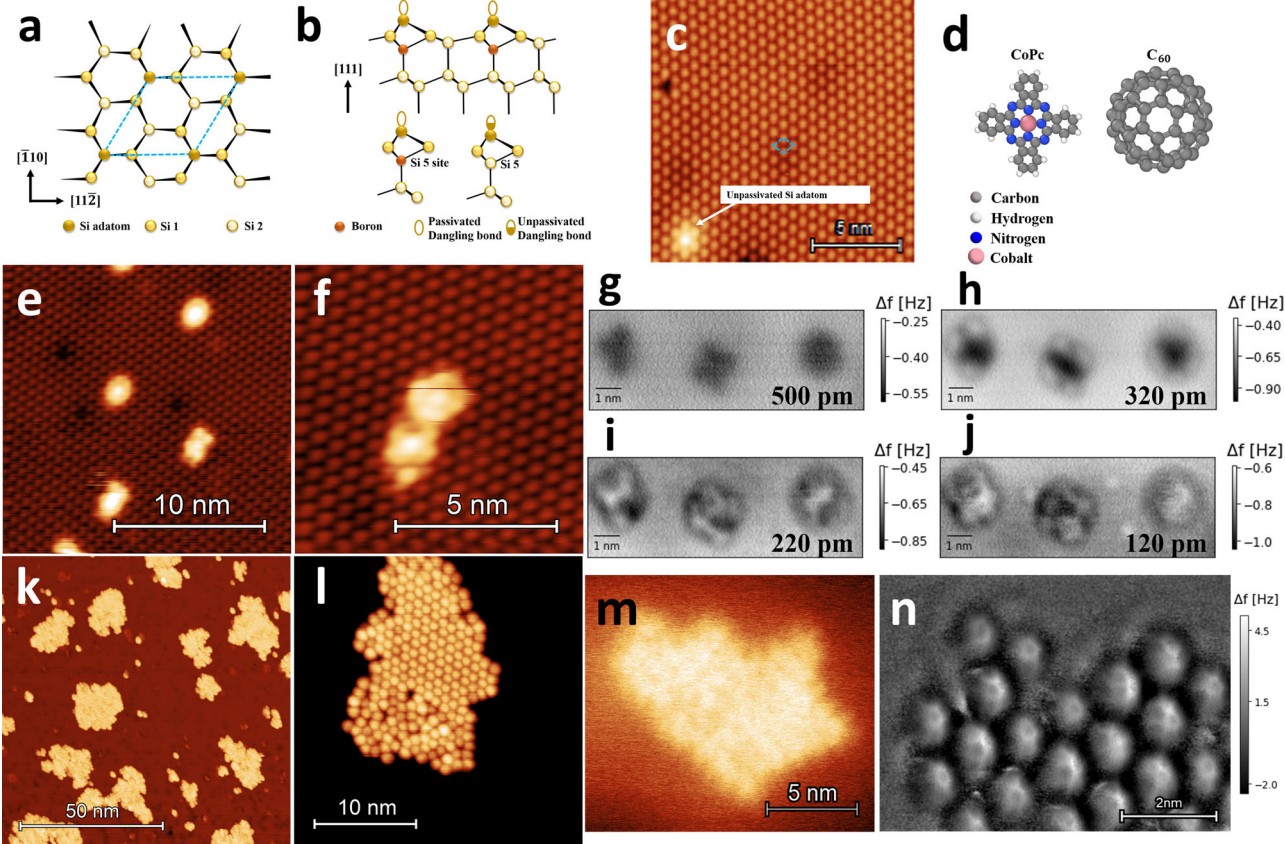

**Fig. 1 Overview of the B:Si(111) substrate, and representative images of the studied molecular systems (CoPc & C$_{60}$) in both STM and NC-AFM. a** Plan view of a ball-and-stick model of the B:Si(111) surface reconstruction. The repeating unit is indicated by the blue dotted line. **b** Top: Side view of the ball-and-stick model of a complete substitution in the B:Si(111) surface. Due to a charge transfer from the silicon adatom to the B atoms, the dangling sp$^3$ bonds of the surface adatoms become saturated[61,62]. Bottom: Depiction of complete and incomplete B substitutions of the S$_5$ site, the latter resulting in surface defects (unpassivated dangling bonds). These defects appear as bright protrusions in STM images, due to their increased electron density[63]. **c** Typical STM image of the B:Si(111) surface, image parameters: $V_{gap} = +2$ V, set point $= 20$ pA. **d** Ball-and-stick models of CoPc & C$_{60}$ (not to scale). **e, f** STM images of CoPc on B:Si(111). Image parameters: $V_{gap} = +2$ V, set point $= 20$ pA. **g–j** Constant height NC-AFM images of CoPc on B:Si(111) acquired at room temperature. Image parameters: $V_{gap} = +0.4$ V, $A_0 = 20$ nm, tracking set point $\Delta f = -1.4$ Hz. Heights relative to tracking position ($\Delta f = -1.4$ Hz set point over the middle molecule): **g** $+500$ pm, **h** $+320$ pm, **i** $+220$ pm, **j** $+120$ pm. **k, l** STM overviews of C$_{60}$ island formation on B:Si(111). Image parameters: $V_{gap} = +2$ V, set point $= 20$ pA. **m** Constant $\Delta f$ NC-AFM overview of a C$_{60}$ island used for subsequent force mapping experiments. Image parameters: $V_{gap} = +0.2$ V, set point $\Delta f = -30$ Hz, $A_0 = 20$ nm. **n** Constant height image of the upper left corner of the C$_{60}$ island. A high pass filter has been applied to highlight the submolecular contrast (which has altered the raw $\Delta f$ values). Image parameters: $V_{gap} = +0.2$ V, $A_0 = 20$ nm, height $= -280$ pm below the tracking position ($\Delta f = -34$ Hz, on a C$_{60}$ molecule outside of frame).

the molecule corners were elucidated across several examples, and compared to DFT simulation. The NC-AFM operation for the islands of C$_{60}$ exhibited particularly high levels of stability, allowing for continuous measurement over a large area of molecules. This stable system lasted for several days, whereby multiple 3D maps were gathered in piece-meal fashion, using both constant height imaging and grid spectroscopy as means of data acquisition. Attempts were also made to study naphthale-netetracarboxylic diimide (NTCDI), a prototypical planar molecule known to form ordered networks on passivated surfaces[25]. However, it was observed that despite the surface passivation, a strong interaction between the NTCDI and the silicon adatoms was still present, inhibiting the formation of ordered molecular networks (see Supplementary Figure 7 in Supplementary Note 5). We note here again, the constraints of the technique, in requiring well-considered molecule-substrate systems for the formation of assembled networks, which will only occur if the molecule-substrate interaction is sufficiently weak. Additionally, for chemical bond resolution, semiconducting substrates are needed to enable tip-passivation.

## Results and Discussion

**Characterisation of isolated CoPc molecules.** A low coverage (<0.1 monolayer (ML)) of CoPc on B:Si(111) was prepared in order to ensure isolated molecules. According to Wagner et al. (2015)[43], the CoPc molecules preferentially pin to the dangling bond defects of the substrate (see Fig. 1c). In constant current imaging in STM, the bright centres of the molecules in Fig. 1e & f indicate the position of the central cobalt core of the molecule, and the four lobes attributed to the corner benzene rings are also resolved. The circular shape of some of the molecules in e) may indicate that they are rotating on the surface. Representative constant height NC-AFM $\Delta f$ images, showing the evolution of the contrast, with varying tip-sample distance, over three molecules with varying adsorption geometries, are depicted in Fig. 1g–j. The middle molecule of the three depicted most closely resembles the expected geometry of CoPc[47]. In images Fig. 1g–i, the lower right benzene ring of the leftmost molecule, exhibits a comparatively weaker tip-sample interaction than the other three corners. The more-positive $\Delta f$ signal indicates that the repulsive features are being probed at these positions. We therefore posit that this

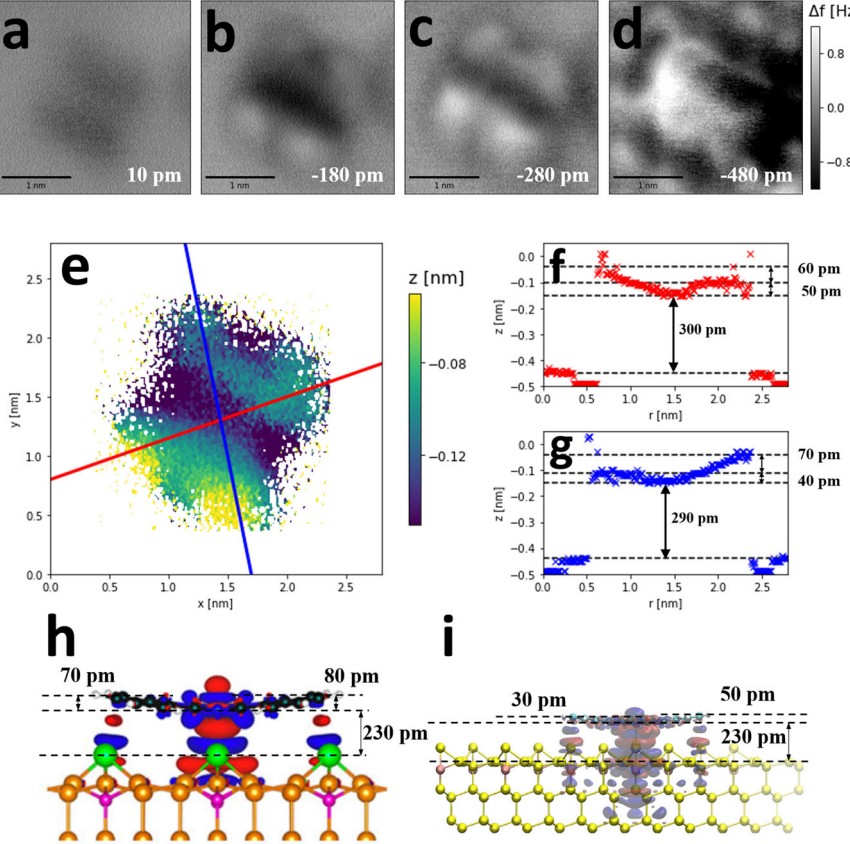

**Fig. 2 3D Δ*f* map of a single CoPc molecule, the contour plots via *z*\*(*x*, *y*) mapping and a comparison to previous DFT study[43,51]. a–d** Representative constant height images of a single CoPc on B:Si(111) taken at varying tip-sample distances in NC-AFM at room temperature (colour scale applies to all four images). **e** Processed *z*\*(*x*, *y*) map of a single CoPc plotted using 3D data of a constant height image set. The complicated tip-sample interactions due to compression / deflection were avoided for the extraction of the *z*\*(*x*, *y*) map, by only considering the first minima reached for each point over the molecule (further details included in Supplementary Note 2). The white pixels indicate the positions within the cube where the turnaround was not reached or was off-molecule. **f & g** Diagonal line profiles across the molecules are plotted for estimation of the molecular distortion. The black dotted lines indicate the relative height of the molecular benzene rings, cobalt core, and the approximate point of Δ*f*(*z*) turnaround on the silicon adatoms (the latter estimated via the on-set of repulsive features over the adatoms, see image **d**). **h & i** DFT calculated structures and charge distributions of cobalt phthalocyanine on B:Si(111), with permission from Wagner et al.[43] and Veiga et al.[51] respectively. **h** Red regions indicate electron accumulation, blue regions indicate electron depletion. i) Blue regions indicate electron accumulation, red regions indicate electron depletion. Estimations of distances were made graphically using the silicon bond length (236 pm) as scale. Dotted lines are added to aid height estimation.

molecule is tilted relative to the surface. This is unlikely to be the result of a tip asymmetry, as this feature is not observed across the other molecules. There is also a difference in the contrast of the cobalt cores of the phthalocyanines. The cores of the left and right molecules exhibit repulsive contrast in images Fig. 1i & j, whereas the core of the middle molecule remains attractive. This likely indicates a variance in cobalt atom protrusion, relative to the phthalocyanine macrocycle. The rightmost molecule has a more circularly symmetric contrast, potentially indicating this molecule is rotating on the surface.

Using a different tip, a 3D Δ*f* map was carried out over a single CoPc molecule via constant height imaging and are plotted in Fig. 2a–d. The constant height images did not reveal the clear, internal chemical structure previously observed with phthalocyanines at low temperature[47,48], or for other molecules at room temperature[30,31], most likely due to a molecular (e.g. CoPc) or multi-atom termination of the tip. Nevertheless, the resolution of the image can clearly resolve differences between different areas of the molecule and the short range Δ*f* contrast in images Fig. 2b, c) shows an apparent tilt in the molecule, indicating the lower left side of the molecule is tilted further up than the upper right side of the molecule. Below a certain tip-sample distance

(approximately 0.3 nm closer to the surface than the tracking position) the contrast evolves in an unusual way, and becomes a more complex convolution of the tip shape and molecule. The tip is likely compressing or deflecting as it approaches the molecule, possibly due to the tip terminating in a CoPc molecule as noted above.

Following a subtraction of a background Δ*f*(*z*) curve[49], the short-range Δ*f*(*x*, *y*, *z*) map was extracted from the raw data and is presented across Fig. 2a–d. The top left corner of the scan frame was taken to be the background signal within the data cube. A benefit of acquiring multidimensional data sets over conventional constant height imaging is that further quantitative details of the system can be extracted. Using the short range Δ*f* cube, a map of the positions in *z* associated with the turnaround in the Δ*f* signal was determined and plotted in Fig. 2e. This is known as a *z*\* map and has previously been used to determine molecular adsorption geometries[11,30], and further chemical information of the system[50]. By selecting the height associated with the Δ*f* turnaround point, the resultant map is effectively a topograph representing the on-set of repulsive forces. Even with a non-ideal tip, the *z*\* data gathered using room temperature NC-AFM reveal that the two benzene rings toward the lower left side protrude

further above the surface than the other two, i.e. there is an apparent tilt across the molecule. The magnitude of the tilt was calculated by extracting the line profiles through the $z^*(x, y)$ data Fig. 2f & g).

The $z^*(x, y)$ map shows the lower left benzene rings of the molecule tilt up above the cobalt core by approximately 110 pm. Conversely, the other two rings tilt up less so, by $\sim 40 - 50$ pm, which is consistent with the $\Delta f$ contrast of the constant height images (see Fig. 1g-j). This level of buckling is slightly greater than that of calculated geometries in DFT studies of CoPc on B:Si(111)[43,51] ($\sim 30 - 80$ pm) (see Fig. 2h & i). This trend is mirrored in Schuler et al.[11], whereby the experimentally determined buckling of a pentacene molecule on a Cu(111) surface is similarly exaggerated by approximately 25 % compared to DFT results. A quantitatively similar exaggeration is observed between the experimentally determined adsorption height of the molecule and DFT studies[43,51]. The experimental data differs from the previous DFT studies in that there is an overall tilt across the molecule, but such a tilt has previously been observed across experimental images of CoPc on an insulating $In_2O_3(111)$ surface at cryogenic temperature[48].

The adsorption height from the DFT calculated structures (shown in Fig. 2h & i) was estimated at 230 pm[43,51], which corresponds to the vertical distance between the centre of the cobalt atom and the silicon adatoms. The $z^*$ measurements indicate a height difference of 290 pm between these atoms, though due to the geometry of the experiment, this corresponds to the vertical distance between the top of the atoms. Accounting for the difference in van der Waals radii for silicon (219 pm) and cobalt (240 pm)[52], this corresponds to a measured adsorption height of 269 pm, which is 17 % greater than the DFT calculated height.

The 3D $\Delta f(x, y, z)$ mapping was repeated over a larger scan area, encompassing a pair of CoPc molecules (with a different tip). Figure 3a-d depict representative constant height images of the scripted set, along with the associated $z^*(x, y)$ map Fig. 3e). Cross-sectional line profiles showing the relative buckling of the molecules are plotted in Fig. 3f-i. For each molecule, there is a

corner / benzene ring that appears to be pulled down toward the surface. Again, this is unlikely to be merely an artefact of tip asymmetry as these particular benzene rings are in different relative positions on the two molecules. For the lower molecule, this relative asymmetry was also present in the constant height images Fig. 3a-d. On the other hand, the upper molecule appears flat in the constant height images, and the buckling is only observed in the $z^*(x, y)$ image. Where the benzene rings tilt up, they do so approximately $70 - 100$ pm above the cobalt core. These values are similar to those of Fig. 2f & g, and are of the same order as the tilt inferred from the DFT results in the literature (Fig. 2h & i). Where the benzene rings do not tilt up, they lie approximately at the same height as the core. Since a $\Delta f(z)$ turnaround was not observed over the substrate adatoms in this data set, an accurate estimate for the adsorption height of the molecule cannot be made.

## Characterisation of molecular islands of $C_{60}$.
The level of deposited coverage was assessed in STM, and is shown inFig. 1k & l. As discussed by Stimpel et al. (2002)[45], the $C_{60}$ initially adsorbs to the defect sites (unsubstituted Si($S_5$)), and along step edges. With sufficient coverage, well-ordered close-packed islands, formed via diffusion, can be observed. Figure 1m & n depict constant $\Delta f$ and constant height overview images of a close-packed $C_{60}$ island in NC-AFM. The non-planar nature of the $C_{60}$ can make resolving the internal structure via constant height imaging challenging. Despite this, clear submolecular features are resolved in each molecule in Fig. 1n. The darker areas depict attractive regions, whereas lighter areas indicate repulsive features, owing to Pauli repulsion, indicating the whereabouts of the chemical bonds of the surface molecules. The structure and shape of these repulsive features in the $\Delta f$ contrast varies from molecule to molecule, suggesting that the $C_{60}$ molecules adsorb in various different orientations. The contrast has similarities to that previously observed on $C_{60}$ molecules in constant height NC-AFM at cryogenic temperatures, for both silicon tips[53] & $C_{60}$ tips[54]. Because of the similarity in image contrast and force minima previously observed for varying types of tip, it is difficult to unambiguously identify the nature of the tip with high confidence, based solely on assessing the image contrast and force map data. However, this does not affect the consideration of the technique as a means of mapping molecular systems.

The island shown in Fig. 1m & n was further studied via multiple 3D $\Delta f$ maps. Figure 4 depicts five $\Delta f$ maps, acquired via constant height imaging in piecemeal fashion. Figure 4a is a single constant height overview image of the island, and serves as a reference image for the subsequent data cubes. We note that acquiring a constant height scan of this size ($15.7$ nm $\times$ 9 nm) at room temperature, requires the tip-sample drift to be highly stable and adequately compensated for. The images Fig. 4c.1-c.4 superimpose the 3D $\Delta f$ data sets, at representative heights, over their approximate corresponding position over the reference image. The thermal drift between the tip and the sample was measured and compensated in between each constant height scan using a few seconds of atom tracking, with sufficient accuracy to afford an acquisition (scan) time of $\sim$3 minutes. A single $C_{60}$ molecule within the island (indicated by the blue $\times$) was used as the tracking site for all five $\Delta f$ maps, hence the tip-sample distances plotted are all relative to the feedback height during tracking associated with the blue $\times$. The drift correction procedure provided long term stability for the room temperature force mapping experiments, continuously over multiple days (the acquisition time of each $\Delta f$ map was $\sim$13 hours). Further details are reported in Supplementary Note 1.

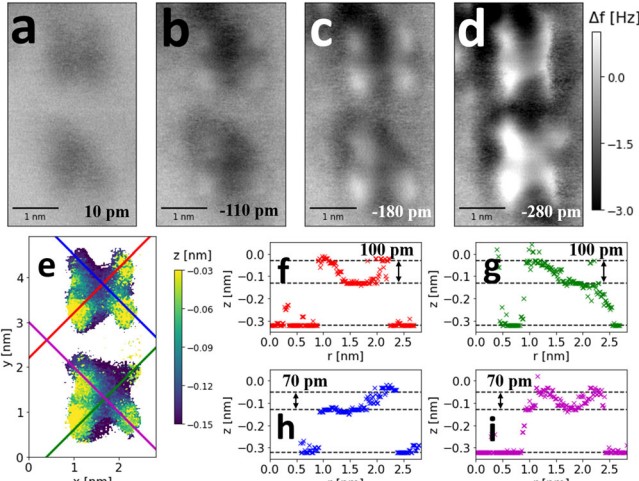

**Fig. 3 3D $\Delta f$ map of a pair of CoPc molecules, and the contour plots via $z^*(x, y)$ mapping. a–d** Representative constant height images of a pair of CoPc molecules on B:Si(111) taken at varying tip-sample distances in NC-AFM at room temperature (colour scale applies to all four images). **e** Processed $z^*(x, y)$ map of CoPc molecules plotted from the 3D constant height image set. **f–i** Diagonal line profiles across the molecules are plotted for estimation of the molecular distortion. The black dotted lines indicate the relative heights of the benzene rings and cobalt core.

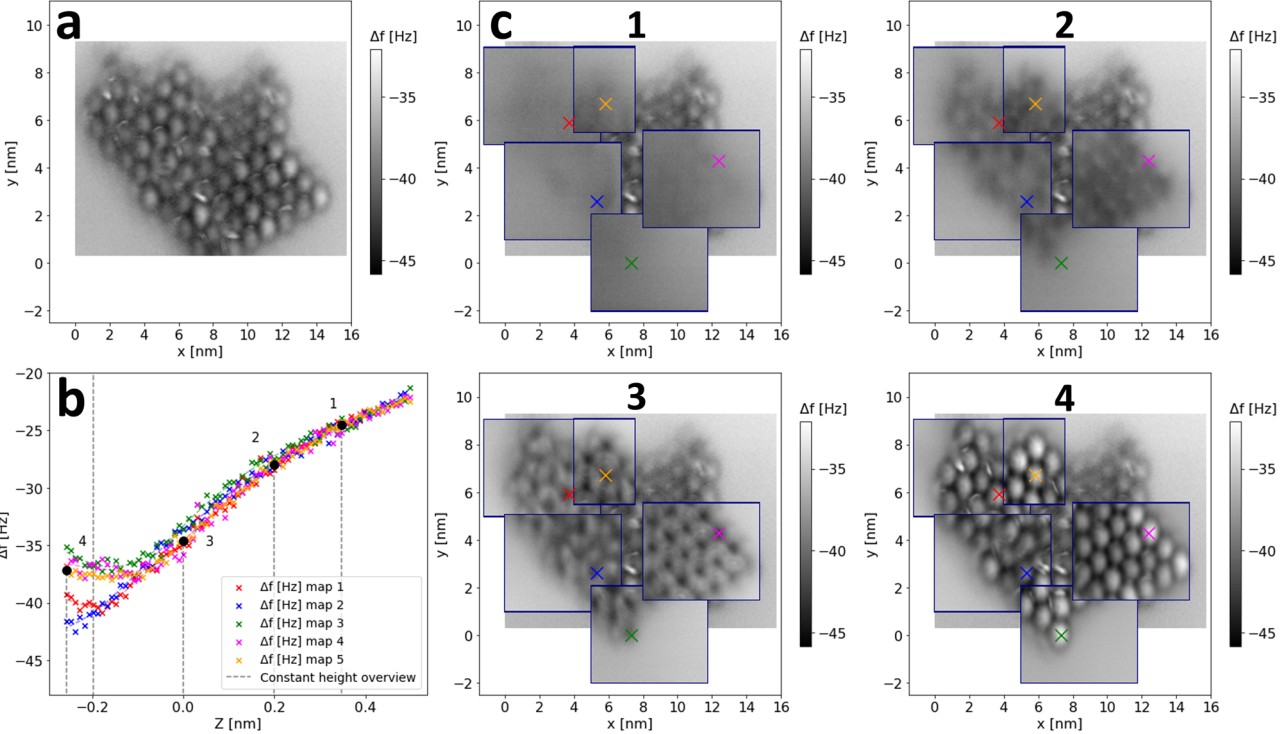

**Fig. 4 Demonstration of a piecemeal method of force mapping, via constant height imaging, across a large scale C$_{60}$ structure. a** Preliminary constant height image of a whole C$_{60}$ island. **b** $\Delta f(z)$ curves extracted from five $\Delta f$ maps. **c** Representative constant height images, across five piecemeal $\Delta f$ maps, superimposed over their approximate corresponding locations within the island in the preliminary scan. Imaging parameters: $V_{gap} = 0.2$ V, $A_0 = 20$ nm, tracking set point $\Delta f = -34$ Hz. All slices in the 3D data sets are averages of three constant height scans. A video version is provided in Supplementary Movie 1.

From each of the five $\Delta f$ maps, a $\Delta f(z)$ curve was extracted over a molecule, aligned in $\Delta f$ (using the same method as described in Brown et al. (2023)[30], and in Supplementary Note 2), and plotted in Fig. 4b. In images Fig. 4c.1-c.4, the positions of each $\Delta f(z)$ curve are indicated by the coloured markers. The shape of the $\Delta f(z)$ curves in Fig. 4b, (and the image contrast) show a systematic change occurred between $\Delta f$ map #2 and map #3. This would indicate the tip had likely changed in some way between these experiments. Nevertheless these data demonstrate the ability of the drift correction procedure to gather multiple $\Delta f(x, y, z)$ maps, at room temperature, over areas large enough to encompass large assemblies of molecules. This process can, in principle, be carried over to more complex self-assembled architectures of molecules.

The $\Delta f(x, y, z)$ data from map #5 (encompassing the orange curve / ×, see Fig. 4) is presented again in Fig. 5a, with further example $\Delta f(z)$ spectra extracted (left), and representative constant height images 1)-4). In Fig. 5a.2 the more negative $\Delta f$ signal over the central positions of the C$_{60}$ molecules indicates a greater attractive interaction than for the background, and exhibits more positive $\Delta f$, indicative of repulsive interactions at the closer tip-sample distances in Fig. 5a.3 & a.4. The areas around the molecule centres show a more positive $\Delta f$ than the background in Fig. 5a.2, but become more negative than the molecules in Fig. 5a.3 & a.4 after the on-set of repulsion for the centres of the molecules. The $\Delta f$ signal of the feature in between the C$_{60}$ (purple ×) exhibits an early on-set of Pauli repulsive interaction, and remains more positive (repulsive) at close tip-sample distances, than the rest of the intermolecular positions. Without a well characterised tip, it is difficult to speculate on the exact mechanism responsible for the repulsive contrast, which was also observed regularly throughout the $\Delta f$ maps and constant height imaging. The tip is again potentially deflecting and compressing at these points, due to an atypical terminating geometry.

The force field over this region was reconstructed by subtracting the background signal of the data set[49,55], and is depicted in Fig. 5b. The force minimum over the C$_{60}$ (red curve) depicts a stronger interaction than observed in previous results for both silicon[53], and C$_{60}$ tips[54], nonetheless the tip-sample interaction is sufficiently weak to allow for imaging in the repulsive regime. The green and purple $F(z)$ curves extracted over the areas in between the molecules show a more complex behaviour toward closer tip-sample distances, likely owing to a non-ideal tip. However, Fig. 5 still demonstrates a proof-of-principle of gathering high-resolution force maps in room temperature NC-AFM in piecemeal fashion.

Following acquisition, another 3D map of the same region was obtained via grid spectroscopy, in order to allow a direct comparison of both methods for the room temperature 3D mapping of molecular samples. Theoretically, both methods of data acquisition should produce equivalent data sets when applied to the same region, as long as the alignment of the data is solely provided in-situ by atom tracking, and does not rely on any post hoc alignment. In practice, experimental limitations can sometimes favour one methodology over another. For example, grid spectroscopy carries the benefit of much more frequent updates of the feedforward correction, as it can pause data acquisition after every $n^{th}$ spectrum. However, grid spectroscopy typically takes significantly longer to gather a full 3D data set, and is more prone to random tip changes, which are likely due to the tip entering the point of closest approach repeatedly for each pixel in $(x, y)$.

This second 3D grid of point $\Delta f(z)$ spectra (acquisition time 16 hours) is presented in Fig. 6a. Using the same subtraction method, the force reconstruction was calculated and plotted in b). The two methods of data acquisition produce 3D maps that are markedly similar. In some instances the systematic error in force

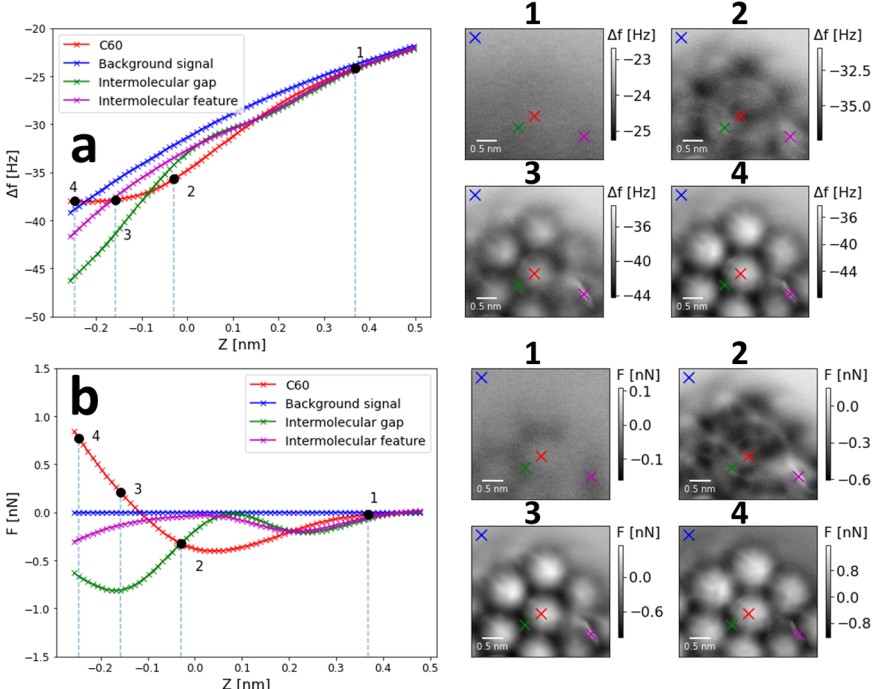

**Fig. 5 $C_{60}$ on B:Si(111) 3D $\Delta f(x, y, z)$ & $F_z(x, y, z)$ maps taken via the constant height imaging method. a** $C_{60}$ on B:Si(111) 3D $\Delta f(x, y, z)$ map taken via the constant height imaging method, map parameters: 352 × 352 × 77 pixels, 3.5 × 3.5 × 0.76 nm³, $V_{gap} = 0.2$ V, $A_0 = 20$ nm, tracking set point $\Delta f = -34$ Hz. The positions of the curves are indicated by the coloured markers in representative image (Right) 1)-4). **b** As for **a**, showing instead the $F_z(x, y, z)$ map calculated using background subtraction to get the short range $\Delta f$ and Sader-Jarvis method of force reconstruction[64]. Details on data processing and force inversion can be found in Supplementary Note 2.

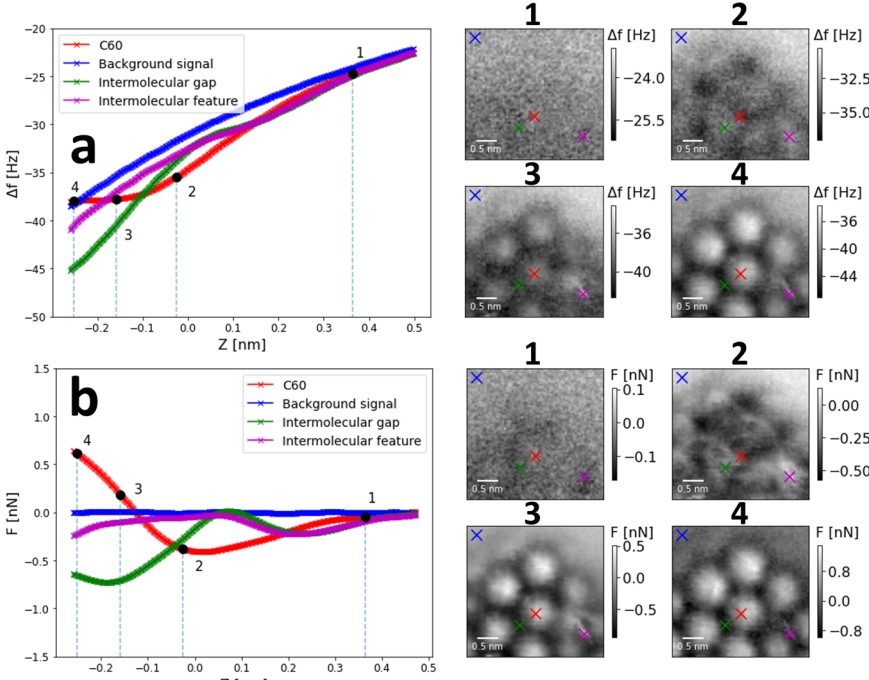

**Fig. 6 $C_{60}$ on B:Si(111) 3D $\Delta f(x, y, z)$ & $F_z(x, y, z)$ maps taken via the grid spectroscopy method. a** $\Delta f(x, y, z)$ map over the same area of $C_{60}$ molecules as for Fig. 5, acquired via grid spectroscopy, map parameters: 70 × 70 × 512 pixels, 3.5 × 3.5 × 0.76 nm³, $V_{gap} = 0.2$ V, $A_0 = 20$ nm, tracking set point $\Delta f = -34$ Hz. **b** As for **a**, showing instead the $F_z(x, y, z)$ map calculated using background subtraction to get the short range $\Delta f$ and Sader-Jarvis method of force reconstruction[64]. Details on data processing and force inversion can be found in Supplementary Note 2.

minima of equivalent sites across the data sets is as small as 10 pN (comparing the force minima from the constant height image and spectroscopy grids, at the site indicated by the red ×, which are −401 pN and −411 pN respectively). However, there are pervasive small quantitative differences between each of the methods. For example, the minima of the tip-sample force over the point indicated by the green × is 90 pN more attractive for the constant height image data set than for the grid spectroscopy method. In addition, the point of closest approach for the red $F(z)$ curve is ~200 pN more repulsive for the constant height imaging method. This is potentially due to the slight differences (<1 Hz) in the respective background $\Delta f(z)$ curves for each data set (the blue curve in a) for each data set, as this would slightly alter the force inversion for all the other $\Delta f(z)$ curves. Alignment of the respective background curves to correct for small thermal $f_0$ drift is not trivial, however, as the data sets have different data densities in 3D space. Further, the respective background $\Delta f$ curves were gathered over differing time spans. For the constant height scan method, it was compiled from $\Delta f$ images gathered across 8 hours, whereas for the grid spectroscopy method, it is a single point spectrum lasting a few seconds, taken upwards of 10 hours later.

From the drift compensation required to keep the tip returning to a constant tracking point, we infer the level of tip-sample drift of the system, and hence, we can compare the stability afforded by each method of data acquisition. We note both are capable of resolving molecular features without tip changes, for durations on the order of days. Empirically, we found constant height scanning a more reliable method compared to grid spectroscopy, which has the tendency to incur a tip change more frequently. Further, while the positional errors of either method are of the order of just a few pm, the drift in the z-axis exhibited higher variation for grid spectroscopy experiments, potentially due to creep effects that come with more frequent movements in z. Despite this, grid spectroscopy still brings utility in submolecular NC-AFM experiments, particularly when used to gather 2D cross-sectional maps, which are comparatively much quicker than 3D experiments[30]. A more detailed comparison of the two methods can be found in Supplementary Note 1.

## Conclusions
Using room temperature NC-AFM we can routinely gather multi-dimensional data sets in constant height over both isolated molecules and large islands of molecules. From these, we observe sub-Angstrom distortions of CoPc molecules on the B:Si(111) surface with comparable magnitude to that estimated from previous DFT studies[43,51], and also to low-temperature studies of CoPc on insulating surfaces[48]. The custom-built drift-correction procedure used for room temperature operation is sufficiently robust and versatile that an island of $C_{60}$ molecules, spanning ~10 × 10 nm$^2$ can be characterised in 3D as well. We note that both constant height imaging and grid spectroscopy are applicable data acquisition methods, which yield a positioning error on the order of pm. The results presented across this paper demonstrate the feasibility of using NC-AFM, in room temperature conditions, to map large and complex molecular systems of interest, in three dimensions and with submolecular contrast. Further, they chart a progression from the mapping of a single, planar molecule[30], helping to bridge the gap between the frontiers of high-resolution imaging, and experiments carried out under conditions closer to real-world application.

## Methodology
STM and NC-AFM data were acquired using a commercial Omicron variable temperature combined STM / NC-AFM. The

microscope was operated via a RC5 Nanonis controller and OC4 phase-locked loop. All experiments were carried out in UHV, at pressures of $\leq 1 \times 10^{-10}$ mbar. For NC-AFM, commercial silicon cantilever tips were cleaned via argon sputtering for 5 minutes (with $1 \times 10^{-5}$ mbar of $Ar^+$ and beam energy 2 kV), which removed the native oxide layer and other contaminant elements, and sharpened the tip. Commercial nanoworld cantilevers, manufactured with spring constant ($k$) 48 Nm$^{-1}$, resonant frequency ($f_0$) ~ 170 kHz, were oscillated at high amplitudes (≥20 nm), and the deflection was measured using a standard four-quadrant photodetector. In a previous study showing room temperature imaging of organic molecules using conventional cantilevers[31], it was noted that the $\Delta f$ signal itself is reduced by the use of large oscillation amplitudes, but the noise density of their deflection sensor was superior to an optimised room temperature qPlus system. In addition conventional silicon cantilevers also garner lower noise contributions from detector, oscillator and thermal sources[56], an estimation of the noise in our instrument is provided in Supplementary Figure 5 (Supplementary Note 3). The oscillation amplitude was estimated using the $\gamma$ method[57], and the level of dissipation in the cantilever is highlighted in Supplementary Figure 6 (and discussed in Supplementary Note 4). The tips were prepared in-situ by gentle indentation into the B:Si(111) surface[31]. A small bias (~0.3 − 0.4 V) was applied to the cantilever tip to remove the electrostatic force due to contact potential difference. At times, a tunnel current was observed in NC-AFM, this is reported in in Supplementary Figure 8 in Supplementary Note 6).

Lateral feedback electronics and a lock-in amplifier were provided in the Nanonis controller atom tracking module. With the tip locked onto a surface feature (silicon adatom or molecule), an average position of the traced path was calculated. This average position was used, in iterative fashion, to calculate and compensate, both the displacement, and drift velocity between the tip and sample. Further details on the procedure are included in in Supplementary Note 1.

**Sample preparation**. The passivated B:Si(111) was formed by using highly boron-doped silicon wafers (0.001 − 0.005 Ω cm resistivity), and thermally activating segregation (the dopants begin to move toward the surface) by annealing the sample at ~ 800 °C. The maximum boron concentration at the surface is 1/3 of a monolayer (1 monolayer being the surface atom density of Si(111): $7.8 \times 10^{14}$ cm$^{-2}$)[58]. Preparation of the B:Si(111) was achieved via flash annealing, cycling between 1200 °C and 800 °C five times, and then leaving to anneal for an hour at 800 °C, before finally slowly cooling the sample to room temperature (<1 °C s$^{-1}$). This procedure would yield an approximate surface defect density of ~5 %. The molecules were deposited from a home-made Knudsen cell onto a room temperature substrate. Isolated CoPc molecules on the B:Si(111) surface were prepared by depositing at 380 °C, for 30 seconds. The sub-monolayer coverage of $C_{60}$ was achieved by depositing at 350 °C, for 30 minutes.

The temperature of the sample was measured and regulated using a Lakeshore temperature controller 331 (via a Pt-100 Ω resistance temperature detector, and a built-in heater at the scan head) during operation of the microscope. The surrounding laboratory temperature was also measured using the Lakeshore controller, and regulated using a standard room air conditioning thermostat. The regulation uses a PID control loop[59], which was manually tuned until a stability of ± 2 mK was observed. Empirically, parameters of $P = 100, I = 10, D = 50$ were found to give the best stability. While stability could be optimised to ± 2 mK, some experiments (used to create the $\Delta f$ maps in Fig. 2 &

Fig. 3) were performed under sub-optimal conditions with a stability of ± 4 mK regulation. However, this did not significantly affect the performance of the drift correction procedure for those experiments.

## Data availability

The raw data presented in the figures within this text and supplementary information are available via doi.org/10.5518/1425[60]. All other data are available on reasonable request.

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

## Acknowledgements

T.B. and A.M.S. thank the Royal Society for funding via a University Research Fellowship and Research Enhancement Award. A.M.S. and P.J.B. thank the ERC for funding under Grant Agreement number 757720 3DMOSHBOND.

## Author contributions

T.B. performed the experiments, analysed the data and wrote the manuscript. A.M.S. conceived the experiments, and provided guidance on the data analysis and manuscript writing. P.J.B. assisted with the experimental set up, data analysis, and manuscript writing.

## Competing interests

The authors declare no competing interests.
