## [Peer Review File · Communications Chemistry]

Reviewers' comments:

Reviewer #1 (Remarks to the Author):

In this paper, Brown et al. report on a protocol for acquiring three-dimensional force map data over organic molecules at room temperature. The authors characterized C60 and phthalocyanine molecules on B:Si(111) using NC-AFM. Getting high resolution AFM images at room temperature is a challenging task. This paper may be useful for developing this technique in the future but I have some concerns.

1) It appears that the main trick to obtain high resolution AFM images is to use the B:Si(111). This is very big limitation, especially to investigate organic layers, because researchers are used to use various substrates to assess intermolecular interactions. This technique is thus not universal.

2) The outcome of this paper is very limited and even superficial in comparison with the authors recent publication (Brown, T., Blowey, P. J., Henry, J. & Sweetman, A. Intramolecular Force Mapping at 469 Room Temperature. ACS Nano 17, 1298–1304 2023).

3) Overall this is a technical paper more than a research paper. Its focuses more on the technique than the new results than this technique can provide.

Overall is not providing more physical insights than the paper recently published by the authors in ACSnano 2023. This technique seems limited to only one type of substrate, which is quite limiting. As it is a very technical paper, I suggest submitting it in a technical/instrumental journal.

Reviewer #2 (Remarks to the Author):

Brown and coworkers explored the application of room-temperature non-contact atomic force microscopy for visualizing cobalt phthalocyanine and C60 molecules on B:Si(111) surface. The authors performed a number of experiments with attaining informative results and demonstrating a new room-temperature method for identifying the molecular adsorption geometry. The manuscript is well organized with ample discussion and comparison with literature. I'd like to recommend its publication in Communications Chemistry after addressing my following concerns.

1. The AFM images of adsorbed CoPc and C60 molecules should be simulated, for comparison and verification.

2. What's the size of the tip apex?

3. What does it look like if the deposition duration for CoPc molecules is extended?

4. I suggest discussing more about the advantages, application range and limitations of room-temperature AFM. For example, the adsorption strength of molecules must be high enough to avoid their movement. This article (Matter, 2021, 4, 4, 1189-1223) would be a good reference.

Reviewer #3 (Remarks to the Author):

The manuscript of "Precise Determination of Molecular Adsorption Geometries by Room Temperature

Non-Contact Atomic Force Microscopy” reports the 3D force mapping on organic molecules using NCAFM at room temperature.

In cryogenic environments, high-resolution observation of individual organic molecules by NCAFM is routinely performed. In many cases, taking advantage of the low-temperature environment, a CO molecule is attached to the tip apex, which is well-defined and sharp, allowing reproducible visualization of the organic molecules. The low temperatures also suppress thermal drift, making it relatively easy to obtain three-dimensional force maps, and studies are being conducted to capture the three-dimensional characteristics of organic molecules.

In this paper, organic molecules (mainly CoPc and C60) adsorbed on semiconductor surfaces are observed by NCAFM at room temperature, and submolecular resolution is obtained. In addition, 3D force maps are obtained and the tilt of the molecules is discussed. The acquisition of 3D force maps on organic molecules at room temperature has been reported previously by the same authors [T. Brown, et al., ACS Nano 17 (2023) 1298], but it is technically very difficult, and the present study, in which data were acquired on different systems of organic molecules, is commendable.

Technically, the protocol for suppressing thermal drift at room temperature is excellent. Also, the signal-to-noise ratio of the frequency shift is well analyzed, and I think this paper will contribute to the future development of this field.

In this study, two methods of obtaining a 3D force map, the slice method and the grid method, are compared. The slice method has been reported in an important previous study [B.J. Albers, et al., Nature Nanotechnology 4 (2009) 307], which is not referred. Although this prior study was done at low temperatures, the drift correction method discussed in that paper should be considered.

Next, comment on the content of the analysis. I would like to see a description of the reasons and motivations for analyzing the organic molecule system used in this study at room temperature. The main result of the analysis is that the organic molecules are tilted from the substrate surface, but it is desirable to discuss the direction of tilting together with the adsorption sites of the organic molecules. The turnaround point (z^*) in the frequency shift curve is used in the discussion, but it should be noted that this value depends on the oscillation amplitude of the cantilever. In the previous study, z^* was analyzed at small amplitudes, which are sensitive to short-range forces, but in this study, z^* is analyzed at large amplitudes, which are sensitive to long-range forces. Since the value z^* is ambiguous, it is better to use a quantity that does not depend on the measurement method, such as the distance between the tip and the sample that gives the maximum attractive force or zero force.

Response to reviewers

Reviewer 1:

In this paper, Brown et al. report on a protocol for acquiring three-dimensional force map data over organic molecules at room temperature. The authors characterized C₆₀ and phthalocyanine molecules on B:Si(111) using NC-AFM. Getting high resolution AFM images at room temperature is a challenging task. This paper may be useful for developing this technique in the future but I have some concerns.

1) It appears that the main trick to obtain high resolution AFM images is to use the B:Si(111). This is very big limitation, especially to investigate organic layers, because researchers are used to use various substrates to assess intermolecular interactions. This technique is thus not universal.

2) The outcome of this paper is very limited and even superficial in comparison with the authors recent publication (Brown, T., Blowey, P. J., Henry, J. & Sweetman, A. Intramolecular Force Mapping at 469 Room Temperature. ACS Nano 17, 1298–1304 2023).

3) Overall this is a technical paper more than a research paper. Its focuses more on the technique than the new results than this technique can provide. Overall is not providing more physical insights than the paper recently published by the authors in ACSnano 2023. This technique seems limited to only one type of substrate, which is quite limiting. As it is a very technical paper, I suggest submitting it in a technical/instrumental journal.

Response:

Our choice of substrate (B:Si(111)) is somewhat arbitrary, and certainly not required for sub-molecular contrast. To be clear, the main constraint of submolecular resolution in NC-AFM at room temperature is the use of a semiconducting substrate, to facilitate tip “passivation” via in-situ preparation on the surface. Sub-molecular contrast is therefore possible on a very wide range of substrates in the absence of CO-tip functionalisation (Si(111) [Nat Commun. 2015, 6 7766], TiO₂(101) [Nano Lett. 2015, 15, 2257–2262], In₂O₃(111) [Surf Sci. 2022, 722, 122065], and Si(111)/CaF₂(111) [Beilstein J. Nanotechnol. 2020, 11, 1432-1438]).

However, in order to facilitate the growth of molecular islands on semiconductors, as opposed to only studying isolated molecules it is often necessary to ‘passivate’ the substrate. B:Si(111) was chosen as it was easy for us to prepare in our chamber which has limited preparation facilities, and this allowed for island formation of C₆₀ molecules. However, there are many other passivated semiconducting substrates such Ag:Si(111) [Nat. Commun, 2014, 5, 3931 & Phys Rev Lett. 2015, 115, 066101] and hydrogen-terminated silicon [Nat. Commun, 2017, 13, 8, 14222], and hydrogen terminated germanium. These surfaces should also be amenable to the same experimental protocols detailed in the manuscript, and indeed current work by the authors pursues these avenues.

Moreover, the drift correction protocols described are applicable to metallic and insulating surfaces, not just semiconductors. The only caveat on operating on these substrates is the difficulty in achieving a stable passivated tip preparation at room temperature.

Importantly, in this manuscript we pay focus to the study of molecular self-assembly with sub-molecular resolution NC-AFM at room temperature. This has not previously been studied at room temperature (only isolated molecules). In particular, a key take-away is the proof of principle demonstration of the drift

correction & force mapping techniques' ability to characterise large, ordered islands in 3D. We feel this is a significant step made beyond our earlier paper [ACS Nano. 2023, 17, 2, 1298–1304], as it is not obvious that the current results would be possible from that paper. Indeed, it is only by significant further development and refinement of our drift correction and temperature stabilisation protocols that these results have been enabled – they were not achievable at the time of the earlier publication.

We also note that although the molecules studied in the manuscript (cobalt phthalocyanine and C₆₀) were chosen owing to their prototypical nature, and the fact that they were already well-studied and characterised throughout the literature. This is because in this paper the methodology employed in operating the NC-AFM was central to the premise of the manuscript. However, we feel the results presented offer a broader scope of possibilities than what this reviewer may have interpreted. We reiterate our above refutation of the idea that large-scale room temperature force mapping is somehow unique to just the single molecule-substrate system presented. Furthermore, we suggest that the ability to characterise 3D nano-architectures across areas the order of 10s of nm (as opposed to just a few nm as previously demonstrated by the authors) at closer to operando conditions, provides a compelling avenue for exploration for functional material synthesis & research, and is not a trivial development from our earlier work.

Reviewer 2:

Brown and coworkers explored the application of room-temperature non-contact atomic force microscopy for visualizing cobalt phthalocyanine and C60 molecules on B:Si(111) surface. The authors performed a number of experiments with attaining informative results and demonstrating a new room-temperature method for identifying the molecular adsorption geometry. The manuscript is well organized with ample discussion and comparison with literature. I'd like to recommend its publication in Communications Chemistry after addressing my following concerns.

- 1. The AFM images of adsorbed CoPc and C60 molecules should be simulated, for comparison and verification.*
- 2. What's the size of the tip apex?*
- 3. What does it look like if the deposition duration for CoPc molecules is extended?*
- 4. I suggest discussing more about the advantages, application range and limitations of room-temperature AFM. For example, the adsorption strength of molecules must be high enough to avoid their movement. This article (Matter, 2021, 4, 4, 1189-1223) would be a good reference.*

Response:

For the CoPc, the DFT studies we reference in the paper [Phys Rev Lett. 2015, 115, 096101 & Phys Rev B, 2016, 93, 115301] carried out by other groups serve to give an approximate sense of the degree to which these molecules lie non-planar to the B:Si(111) surface, and give some theoretical backing to the variation in contrast we observe across different CoPc and the relative buckling of their benzene rings in our constant height images contour plot. Moreover, we do not currently have the ability to carry out in-house calculations of the different molecular geometries via DFT. A full DFT investigation of the different

adsorption positions of CoPc on B:Si(111), which would be required to produce PPM based simulated images, is beyond the scope of this paper, which is more focused in demonstrating a proof-of-principle of the force mapping techniques in concert with the drift correction procedure at room temperature.

For the C₆₀, we feel simulations are unlikely to elucidate any further information of the molecule-substrate system. This is owing to the lack of unambiguous resolution in our constant height Δf data, which could result from either C₆₀-terminated or silicon-terminating tips [Nat Commun. 2016, 7, 10621 & Phys Rev B. 2016, 94, 115440]. Without additional characterisation of the probe apex it would be almost impossible to perform simulations that would reliably reproduce the experimental result

Due to the current lack of reliable inverse imaging protocol on this substrate [e.g. Phys Rev Lett. 2012, 108, 268302], we are unable to determine the exact termination of our tips. However, more broadly we can suggest that our tip is terminated by a single atom or molecule at the apex, as evidenced by the routine atomic and submolecular resolution they provide. If, however, this reviewer's question was pertaining to the nanoscopic size of the tip (rather than the apex termination), this is again difficult to accurately quantify in-situ. We employ commercial Nanoworld silicon cantilevers, whose nanoscopic tip radius is likely to be approximately 10 -20 nm.

We refer to a paper in the manuscript text wherein further deposition of CoPc has been demonstrated [Phys. Rev. Lett., 2015, 115, 096101]. The outcome is disordered adsorption, until coverage is sufficient that non-planar $\pi - \pi$ stacking starts to occur between the molecules in the same way as for Cu and Zn phthalocyanines. As we required semiconducting silicon atoms to be accessible to prepare our tips, we favoured a minimal deposition for the CoPc.

The reviewer is correct in that the technique described in the manuscript is reliant on an appropriately strong adsorption of the molecule on the substrate, as to avoid diffusion during data acquisition which is more prevalent in general at room temperature. We agree it is worth noting more explicitly in the text, when the technique will and won't be applicable i.e. the tip also requires passivation, terminating with either a larger molecule or semiconducting cluster to achieve chemical-bond resolution. We have amended the manuscript to make this point clearer.

Manuscript lines: 85-91

"As such, our substrate needed to be semiconducting, as these can facilitate, via gentle indentation into the surface, the chemical passivation of the tip [Nat Comm. 2015, 6, 7766]. Preparation of the tip in this way can yield a molecular or semiconducting cluster termination, which can be conducive to unreactive tips and hence, chemical-bond resolution []. Concurrently, the molecule-substrate interaction must be sufficiently strong to inhibit molecular diffusion during imaging (which is prevalent in room temperature conditions), however it also ought to be weak enough that some amount of diffusion can occur in order to form intermolecular bonds when appropriate [Matter. 2021, 4, 1189-1223]. The Si(111)-(7X7) reconstructed surface, as used in the aforementioned work by the current authors. . . "

Manuscript lines: 123-128

"it was observed that despite the surface passivation, a strong interaction between the NTCDI and the silicon adatoms was still present, inhibiting the formation of ordered molecular networks (see Supplementary Section 4). We note here again, the constraints of the technique, in requiring well-considered molecule-substrate systems if the user requires the formation of assembled networks. The molecules will only form intermolecular bonded networks if their interaction with the substrate is sufficiently weak. For chemical bond resolution, semiconducting substrate are necessary to enable tip-passivation."

Reviewer 3:

The manuscript of "Precise Determination of Molecular Adsorption Geometries by Room Temperature Non-Contact Atomic Force Microscopy" reports the 3D force mapping on organic molecules using NCAFM at room temperature.

In cryogenic environments, high-resolution observation of individual organic molecules by NCAFM is routinely performed. In many cases, taking advantage of the low-temperature environment, a CO molecule is attached to the tip apex, which is well-defined and sharp, allowing reproducible visualization of the organic molecules. The low temperatures also suppress thermal drift, making it relatively easy to obtain three-dimensional force maps, and studies are being conducted to capture the three-dimensional characteristics of organic molecules.

In this paper, organic molecules (mainly CoPc and C60) adsorbed on semiconductor surfaces are observed by NCAFM at room temperature, and submolecular resolution is obtained. In addition, 3D force maps are obtained and the tilt of the molecules is discussed. The acquisition of 3D force maps on organic molecules at room temperature has been reported previously by the same authors [T. Brown, et al., ACS Nano 17 (2023) 1298], but it is technically very difficult, and the present study, in which data were acquired on different systems of organic molecules, is commendable.

Technically, the protocol for suppressing thermal drift at room temperature is excellent. Also, the signal-to-noise ratio of the frequency shift is well analyzed, and I think this paper will contribute to the future development of this field.

In this study, two methods of obtaining a 3D force map, the slice method and the grid method, are compared. The slice method has been reported in an important previous study [B.J. Albers, et al., Nature Nanotechnology 4 (2009) 307], which is not referred. Although this prior study was done at low temperatures, the drift correction method discussed in that paper should be considered.

Next, comment on the content of the analysis. I would like to see a description of the reasons and motivations for analyzing the organic molecule system used in this study at room temperature.

The main result of the analysis is that the organic molecules are tilted from the substrate surface, but it is desirable to discuss the direction of tilting together with the adsorption sites of the organic molecules.

The turnaround point (z^*) in the frequency shift curve is used in the discussion, but it should be noted that this value depends on the oscillation amplitude of the cantilever. In the previous study, z^* was analyzed at small amplitudes, which are sensitive to short-range forces, but in this study, z^* is analyzed at large amplitudes, which are sensitive to long-range forces. Since the value z^* is ambiguous, it is better to use a quantity that does not depend on the measurement method, such as the distance between the tip and the sample that gives the maximum attractive force or zero force.

Minor:

Page 10, line 166

data Fig. 2F & D

should be

data Fig. 2F & G

Response:

- 1) It is unwise to apply a post-hoc correction method as suggested by Albers et al. , without judicious knowledge of the evolution of the tip-sample junction, or a known reference point in the image. The reason being at close approach it becomes difficult to unambiguously determine when movement of features are artefacts due to drift, or owing to genuine tip-sample interaction. Fitting the data to a pre-conceived idea what the system is expected to look like brings risk of**

'correcting for' features that actually arise owing to physical properties of the tip e.g. [Phys Rev B. 2011, 83, 035421]. To emphasize the importance of the real time drift correction used in our work we have added this reference and the following text to the paper:

Manuscript lines: 69-72

"We favoured an in-situ method of drift correction as opposed to post-*hoc* alignment methods [Nat Nano. 2009, 4, 307-310], to better ensure that the molecular features imaged at close approach arose from genuine tip-sample interactions rather than potential artefacts of data correction [Phys Rev B. 2011, 83, 035421]."

We thank the reviewer for highlighting the importance of this point in the manuscript to our attention.

- 2) The precise adsorption geometry is studied in detail theoretically in the papers [Phys Rev Lett. 2015, 115, 096101 & Phys Rev B, 2016, 93, 115301] which are referred to in the manuscript (DFT calculations). In particular [Phys Rev Lett. 2015, 115, 096101] presents three stable configurations of the CoPc orientation relative to the lattice, owing to its particular orbital-hybridization mechanism with the B:Si(111). The central premise of our manuscript is a proof-of-concept of the techniques and protocols employed to correct for thermal drift over larger areas and longer data acquisition durations than previously reported for room temperature NC-AFM. As such, assembling the necessary experimental and simulated data to systematically investigate the molecule-substrate interactions described in these referenced works is secondary to assessing the merits of the methodology. Both molecular-substrate systems were chosen by virtue of their well-studied and prototypical nature. CoPc was chosen because, unlike other phthalocyanines, it lies flat on the B:Si(111) (as discussed in the introduction). Conversely, C₆₀ was chosen owing to its island formation on the surface [Materials Science and Engineering B. 2002, 89, 394-398].
- 3) Whether we extract the topographic (or z*) maps from the turnaround point of the Δf signal or from the force does not broadly affect the results. We elected (both in the manuscript and in our previous work [ACS Nano. 2023, 17, 2, 1298–1304]) to use the turnaround point in the Δf signal for determining the z* maps for simplicity and also reduced noise in comparison to the force signal.
- 4) We thank this reviewer for their careful reading of the paper and have amended the error in reference to the in-paper figure.

REVIEWERS' COMMENTS:

Reviewer #3 (Remarks to the Author):

I have reviewed the responses to the comments of all referees and the revised manuscript.
I recommend publication in Communications Chemistry.